# Impact of Modern Lifestyle on Circadian Health and Its Contribution to Adipogenesis and Cancer Risk

**DOI:** 10.3390/cancers16213706

**Published:** 2024-11-01

**Authors:** Oxana Dobrovinskaya, Javier Alamilla, Miguel Olivas-Aguirre

**Affiliations:** 1Laboratory of Immunobiology and Ionic Transport Regulation, University Center for Biomedical Research, University of Colima, Colima 28040, Mexico; oxana@ucol.mx; 2Consejo Nacional de Humanidades, Ciencia y Tecnología (CONAHCYT), Programa de Investigadores e Investigadoras por México, México City 03940, Mexico; javier_alamilla@ucol.mx; 3Centro Universitario de Investigaciones Biomédicas (CUIB), Universidad de Colima, Colima 28040, Mexico; 4Laboratory of Cancer Pathophysiology, University Center for Biomedical Research, University of Colima, Colima 28040, Mexico

**Keywords:** circadian clock, cortisol, obesity, cancer, chemoresistance, melatonin, tumor microenvironment

## Abstract

This research explores the impact of modern lifestyle factors, such as exposure to artificial light, shift work, and dietary habits, on our body’s internal clock and its association with increased body fat deposits and cancer risk. By reviewing various published studies, we aim to understand the physiological mechanisms underlying the increased risk of cancer associated with an unhealthy lifestyle. In particular, there is evidence that disruptions in the body’s natural rhythms can lead to abnormal cortisol levels, which in turn may promote fat buildup and create an adipose environment favorable for cancer growth. Additionally, we explore the anticancer effects of melatonin, either directly on cancer cells or through its role in preventing the formation of an adipose microenvironment conducive to tumor development. Our findings could offer new insights into how lifestyle choices and melatonin regulation influence cancer development, helping researchers and healthcare professionals better understand and address these risks.

## 1. Introduction

In recent years, there has been a growing interest in elucidating how the regulation of circadian rhythms preserves individual health. Research has increasingly focused on how disruptions in circadian rhythms may contribute to various health conditions, including mental disorders, metabolic diseases, and cardiovascular diseases, among others [1,2,3]. For instance, studies have shown that individuals with irregular circadian patterns are at higher risk for certain cancers, such as breast and prostate cancer [4,5]. However, these associations remain controversial for various cancer types, and the pathophysiological mechanisms underlying such observations have not been thoroughly understood.

Circadian cycles are under the control of the suprachiasmatic nucleus (SCN), considered the master clock. While the SCN maintains rhythmicity through clock gene regulation, light exposure remains a principal synchronizer for optimal central clock function [6,7]. Beyond light, factors such as body temperature, nutrition, hormones, and daily habits also influence circadian rhythms. The present study aims to investigate how modern lifestyle factors—such as non-natural light exposure, shift work, and dietary patterns—disrupt circadian rhythms, focusing on hormonal influences, particularly cortisol and melatonin. The issue of our particular interest is how these alterations lead to adipocyte accumulation in tissues, giving that an adipocyte-enriched tumor microenvironment (TME) may facilitate cancer development.

In this review, we adopted a narrative approach to examine the literature on circa-dian disruption, adipogenesis, and cancer development, rather than employing a system-atic review methodology. The selection of articles was based on the relevance of their contributions and varied depending on the chapter being developed, with the specific criteria outlined as follows. In Chapter 2, the selection criteria were not overly strict; we excluded novel articles that merely recapitulate or repeat the findings of foundational studies. Instead, only original articles that experimentally establish the fundamental principles of circadian rhythms were considered. In Chapter 3, the selection criteria were considered the most rigorous. The first criterion focused on studies evaluating modern lifestyle habits that have the potential to disrupt circadian cycles. Search terms included “shift work”, “insomnia”, “irregular sleep patterns”, “non-natural light exposure”, “light at night”, “artificial light”, “dietary habits”, “fasting”, “alcohol consumption”, “caffeine intake”, and “sweetened drinks”, using databases such as Scopus, Google Scholar, and PubMed. The second criterion involved the authors’ judgment on the relevance and pertinence of the studies, specifically targeting articles documenting quantitative alterations in cortisol or melatonin chronobiology, such as changes in peak levels, baseline variations, new peak formations, phase shifts, or suppression. Articles that did not quantitatively document these changes or that reported them without adequate data were omitted. Additionally, the search results for some modern habits were excluded due to divergent findings and a lack of consensus in the results, ensuring that the focus of the research remained clear. In Chapter 4, a narrative synthesis of original research and reviews that clearly outline adipogenesis and the tumor microenvironment was conducted. Articles supporting the proposed new perspective or hypothesis were selected, critically analyzing the effects of the circadian disruptions mentioned in the previous chapters on adipogenesis. This allowed for an objective interpretation of the interrelationship between circadian cycles, obesity, and cancer, which remains an underexplored area. Finally, in Chapter 5, the same criteria as in Chapter 3 were applied, focusing on scientific articles documenting melatonin dysregulation in chronobiology. Inclusion criteria encompassed studies that report alterations in melatonin production, its anticancer effects (either directly or as an adjuvant), or its role in promoting an adipose microenvironment. 

By clarifying the direct relationship between circadian disruption, adipose microenvironment promotion, and the formation of cancer cell sanctuaries, this research seeks to address existing gaps in the understanding of circadian health and cancer promotion, representing a significant advancement in the field.

## 2. Fundamentals of Circadian Rhythms 

### 2.1. Circadian Rhythms and Their Central Regulation in Mammals 

Circadian rhythms are biological processes with an approximate 24-h periodicity, found across nearly all organisms, from single-celled bacteria to humans. The term “circadian”, derived from the Latin *circa* (“around”) and *diem* (“day”), was first introduced by Franz Halberg in 1950 [8]. These rhythms regulate numerous behavioral, physiological, and biochemical processes, including hormonal secretion, sleep–wake cycles, body temperature, and heart rate [9,10].

In mammals, the suprachiasmatic nucleus (SCN), located in the anteroventral hypothalamus above the optic chiasm, serves as the central circadian regulator. Each SCN, composed of approximately 10,000 neurons in rodents, is organized into a ventral “core” and a dorsal “shell”. The core, which receives direct input from the retina, is distinguished by the production of vasoactive intestinal peptide (VIP), neurotensin, and gastrin-releasing peptide (GRP), while the shell primarily expresses arginine vasopressin (AVP) [11]. Most SCN neurons are GABAergic, though a subset is glutamatergic, and both neurotransmitters are co-localized with the aforementioned peptides [12].

Circadian rhythms are endogenous but can be synchronized or entrained to external cues (zeitgebers), with light being the most prominent. The retino-hypothalamic tract (RHT) connects intrinsically photosensitive retinal ganglion cells (ipRGCs) to the SCN, mediating light-induced entrainment. These ipRGCs, expressing the photopigment melanopsin, send signals to the SCN, intergeniculate leaflet, and other brain regions responsible for circadian regulation, the pupillary light reflex, and mood [13,14]. The RHT releases glutamate (Glu) and pituitary adenylate cyclase-activating polypeptide (PACAP) at synapses in the SCN, activating the N-methyl-D-aspartate (NMDA) and α-amino-3-hydroxy-5-methyl-4-isoxazolepropionic acid AMPA/kainate receptors [15]. This process leads to membrane depolarization and subsequent Ca^2+^ influx, triggering signaling pathways, including the Ca^2+^/Calmodulin-dependent protein kinase (CaM kinase), the cAMP/protein Kinase A PKA, and the ERK/MAPK pathway, as well as the expression of *CREB*, *Per1* and *Per2* genes, which are necessary for phase advances and delays essential for adjusting the circadian clock to the light [16,17,18,19,20,21,22,23,24,25]. This intricate system demonstrates how circadian rhythms in mammals are both self-sustained and responsive to environmental cues, particularly light, which plays a critical role in the synchronization of biological processes to the external world. 

### 2.2. Mechanism and Outputs of the Molecular Circadian Clock

The molecular circadian clock operates through a transcriptional feedback loop governed by “clock genes”. Central to this mechanism are the *Clock* and *Bmal1* genes, encoding the CLOCK and BMAL1 proteins, which form heterodimers that activate the transcription of target genes by binding to E-box DNA elements [26,27]. Among the activated genes are *Per* (*Per1*, *Per2*) and *Cry* (*Cry1*, *Cry2*), which form the negative arm of the feedback loop. The resulting PER and CRY proteins also form heterodimers that inhibit the transcriptional activity of CLOCK:BMAL1, thus completing the feedback cycle [28,29,30]. Furthermore, the clock genes modulate epigenetic processes such as histone acetylation and deacetylation, notably of H3 and H4, which are crucial in maintaining transcriptional regulation and determining the phase of the clock [31,32].

The SCN, known as the central pacemaker, regulates several key circadian outputs. These include the behavioral sleep/wake cycle and hormone secretion patterns, such as adrenal corticosterone and melatonin production in the pineal gland through a multisynaptic pathway involving various regions, including the paraventricular nucleus of the hypothalamus, preganglionic sympathetic neurons, and the superior cervical ganglion [33,34,35,36]. The SCN’s regulation of melatonin exhibits a day–night pattern, exerting inhibitory control during the day and promoting release at night [37]. Beyond hormonal regulation, other neurocrine factors released by the SCN, such as the Transforming Growth Factor alpha, TGF-α, prokineticin-2, and neuromedin S, influence locomotor and anorexigenic signaling [38,39,40,41]. Furthermore, SCN projections to various hypothalamic regions, including the medial preoptic area, dorsomedial nucleus, and subparaventricular nucleus, regulate neurons controlling hormone secretion such as corticotropin-releasing hormone (CRH) and gonadotropin-releasing hormone (GnRH) [42,43,44,45]. Notably, the SCN regulates the circadian rhythm of corticosterone via the hypothalamo–pituitary–adrenal (HPA) axis through mechanisms involving vasopressin and other neurotransmitters, which modulate the activity of the paraventricular nucleus and downstream adrenal cortex, ultimately leading to the timed release of cortisol. [44,46,47,48].

Considering the influence of SCN activity on multiple physiological processes, it is essential to maintain its proper regulation to ensure optimal health. Disruptions in SCN signaling can lead to dysregulated cortisol production, which has been linked to metabolic disorders, impaired immune function, and increased stress susceptibility. Given the critical role of the SCN in coordinating circadian rhythms, modern lifestyles can significantly disrupt circadian health. These disturbances warrant further investigation to understand their long-term impact on physiological systems and to develop strategies to mitigate their detrimental effects on circadian regulation and overall health.

While this chapter has focused on providing an overview of the original works that establish the fundamentals of SCN function and the generation of circadian rhythms, we acknowledge the existence of excellent recent reviews in the field of circadian biology that are recommended for further exploration.

## 3. Impact of Modern Lifestyle on Cortisol Production and Predisposition to Adipogenesis

In this Section, we aim to explore how various aspects of modern lifestyles, such as daily exposure to light sources, shift work, insomnia, and dietary habits, may contribute to the dysregulation of circadian rhythms to promote adipogenesis and accumulation of adipocytes in tissues. Although there are multiple parameters for assessing circadian disruption—such as behavioral measures (sleep–wake cycles, actigraphy), physical parameters (temperature variations), genetic assessments (quantification of clock gene expression), and molecular markers (estimation of hormones like cortisol and melatonin)—this section mainly focuses on molecular changes. Cortisol is an ideal marker for assessing circadian health due to its characteristic rhythmic secretion pattern. In humans, cortisol levels peak in the early morning and decline throughout the day. This predictable pattern directly reflects the activity of the central biological clock (SCN) and is critical for regulating key physiological processes such as metabolism, immune response, and stress adaptation. Additionally, the evaluation of cortisol is minimally invasive, as it typically requires only saliva or blood samples, making it convenient for regular monitoring. 

### 3.1. Deregulation of the Circadian Cycles and Predisposition to Adipogenesis 

As described in Chapter 2, SCN is recognized as the principal biological pacemaker. It orchestrates rhythmic activities and behaviors across various cells, including those in peripheral tissues, through both neuronal projections and hormonal signals, primarily affecting the release of melatonin and cortisol. The SCN’s influence is mediated by its intrinsic activity and its response to external stimuli, such as light–dark cycles [49]. By integrating internal rhythms with environmental signals, the SCN coordinates systemic responses, such as cortisol release, to optimize the organism’s functioning according to its activity period—diurnal in humans and nocturnal in rodents.

Modern lifestyles, however, introduce activities and factors that can disrupt SCN function and alter systemic signals, potentially leading to pathological states. These disruptions include changes in sleep patterns due to shift work, night shifts, or voluntary sleep deprivation related to social activities. Furthermore, exposure to either dim or bright external light, particularly blue light, at irregular times—whether prolonged or nocturnal—can exacerbate these disturbances. Dietary factors such as prolonged fasting, high carbohydrate meals, or alcohol/caffeine consumption may also contribute to SCN dysregulation and impaired cortisol production (see Appendix A). The next section will provide evidence of how contemporary habits disrupt circadian cycles at the hormonal level and explore their implications for adipogenesis and obesity.

#### 3.1.1. Irregular Sleep Patterns

To maintain mental and physical health, it is crucial to have adequate sleep episodes, which may fluctuate throughout life but generally range from 7 to 11 h per day [50]. During this time, the organism experiences changes in neural, metabolic, thermal, and hormonal patterns necessary for its restoration. However, sleep deprivation—whether due to social or professional reasons, partial or total, acute or chronic—results in significant alterations in cortisol release, which disrupt overall bodily function (Appendix A). Such alterations are evident as elevated baseline cortisol levels and excessive cortisol production (a 37% increase in partial deprivation and up to 45% in total deprivation) and modifications in release kinetics (e.g., delaying cortisol release by hours on subsequent days) [51,52,53]. Additionally, cortisol peaks are present at unusual times, such as during the night, for humans [54].

Shift workers, such as nurses, security guards, medical staff, and those with frequent schedule changes, like pilots, have been commonly studied to assess the impact of altered sleep patterns on health. Indeed, the International Agency for Research on Cancer (IARC), a specialized agency of the World Health Organization, has recognized shift work as a carcinogenic risk to humans [55]. These and other recent studies clearly indicate that both sleep deprivation and changes in sleep patterns cause significant disruptions in cortisol production, increasing it and prolonging its duration [51,52,53,54,56,57,58,59]. These alterations in cortisol production are further heightened when sleep deprivation is coupled with operational tasks, which is common for many professionals [60]. While the entire population is subject to sleep disturbances, these unhealthy patterns remain a significant occupational issue, particularly for healthcare workers. Moreover, events like the recent pandemic have exacerbated this problem, leading to notable sleep disorders in up to 41% of healthcare workers [61,62]. Furthermore, the general population experienced delays in bedtime, increased time in bed, and longer sleep duration due to the lockdown; however, reported sleep efficiency decreased. Additionally, the use of screen media and electronic devices increased and was associated with poorer sleep quality [63]. While there is substantial scientific evidence linking sleep restriction to weight gain and obesity, it has been concluded that the mechanistic relationship remains unclear. [64]. In this context, it has been observed that patients with sleep disorders, such as obstructive sleep apnea, exhibit alterations in the circulating profile of microRNAs. Specifically, there is an increase in the production and secretion of miR-181a, which, through clock genes (PER3), promotes the differentiation of MSCs into adipocytes [65,66].

#### 3.1.2. Non-Natural Light Exposure 

Artificial lighting plays a crucial role in contemporary life by allowing us to extend our activities into the evening and nighttime, thereby optimizing productivity and flexibility. It enhances safety and comfort in our daily environments, supporting everything from household tasks to professional work, and contributes to an overall improved quality of life by ensuring that essential functions can continue regardless of natural light availability.

The increasing prevalence of mobile devices, computers, televisions, and tablets has made artificial light sources more ubiquitous, affecting people of all ages, including infants and young children who are exposed to screens for entertainment purposes. Despite this widespread use, there are no international regulations governing light exposure. However, substantial evidence indicates that various types of artificial lighting can have significant impacts on human health, potentially leading to pathological alterations.

We have previously established that the regulation of SCN activity is primarily influenced by light stimuli (Chapter 2 [6,67]). Nevertheless, the potential impact of excessive use of artificial light sources on human health, particularly on human physiology and behavior, has been underestimated to date. 

Exposure to artificial light is detected by the ipRGCs, and through the RHT transmit inhibitory signals to the SCN, leading to the release of neurotransmitters such as GABA and neuropeptides that reduce SCN neuronal activity, ultimately decreasing melatonin secretion from the pineal gland and disrupting the overall circadian rhythm [68,69].

Through the analysis of global satellite imagery paired with worldwide trends in obesity and overweight, it has been established that the amount of nocturnal artificial light (ALAN) is a predictor for developing overweight and obesity within illuminated communities over the next decade [70,71]. Similar findings have been observed for indoor lighting, such as in bedrooms, where the intensity of light is directly associated with the BMI of the individuals residing in these spaces [72]. While most studies have fallen short in mechanistically explaining the association between ALAN and overweight or obesity, many attribute these observations to dysregulation in melatonin production and changes in feeding patterns during light exposure. However, some research has found that these associations are independent of melatonin production and suggests that additional alterations could contribute to these observations, warranting further detailed exploration [72].

In this context, we collected data from independent studies worldwide, demonstrating that stimulation with artificial light sources, not only nocturnally but also after sunrise, effectively contributes to the dysregulation of melatonin, alterations in clock gene expression, and, importantly, the dysregulation of cortisol production (Appendix A). These findings lead to precise conclusions, such as acute exposure to intense white light increases cortisol production by up to 54% in healthy individuals [73]. Additionally, specific trials have determined that the influence of light on cortisol production is preferentially promoted by light spectra near blue (407 nm) and green (520 nm) wavelengths [73,74,75,76,77,78].

It is important to note that the associations between the quantity and distribution of artificial light not only correlate with the propensity to develop obesity but also significantly correspond with the risk distribution for cancer, as observed in independent studies [79,80,81]. This can be partly explained by the fact that cortisol is the most well-known endogenous immunosuppressant, and alterations in its production due to exposure to intense or blue/green ALAN can directly contribute to cancer promotion by depressing the immune system of an organism (e.g., the anticancer activity of lymphocytes). However, as will be reviewed in Chapter 4, the consequences of dysregulated cortisol can impact other levels to favor cancer progression.

#### 3.1.3. Dietary Habits

Modifications in dietary patterns, including various forms of fasting, such as chronic, partial, and intermittent fasting, have become prevalent trends. While substantial evidence supports the metabolic benefits of these dietary regimens—such as improvements in glucose levels, mental function, and cardiovascular health—concerns about potential adverse effects also arise. Adaptations to fasting include changes in circadian hormone behaviors, such as increased cortisol levels and decreased melatonin levels [82,83]. A systematic review and meta-analysis of 13 independent studies revealed that total caloric restriction due to fasting significantly increases cortisol levels. However, this effect was not observed with severe or mild caloric restriction but was specific to fasting [84,85]. Recently, Lowell’s group explained that fasting triggers survival adaptations, including the activation of the HPA axis and subsequent release of cortisol [86]. Although this activation is a response to the uncertainty of future food intake and aims to mobilize available energy sources, elevated cortisol levels might also impact other physiological processes and potentially have counterproductive effects.

Alcohol consumption, though often normalized as a common social and recreational activity, has significant physiological impacts, especially when consumed in excess. Experimental studies involving ethanol intoxication in individuals with alcoholism have demonstrated a positive correlation between blood alcohol levels and cortisol levels [87,88]. Cortisol levels can increase by approximately 0.18 micrograms/dL for each alcoholic drink consumed per day (about 14 grams of alcohol) [89]. Furthermore, cortisol levels remain elevated in individuals after alcohol withdrawal compared to abstinent individuals [90]. Basal cortisol levels are also higher in individuals with a family history of alcoholism after consuming alcohol compared to those without such a family history]. Additionally, alcohol abuse disrupts sleep, exacerbating issues such as sleep apnea, insomnia, and poor sleep quality, which can further aggravate systemic cortisol responses [91,92,93].

Caffeine, a widely consumed substance, regulates the cortisol response through its effects on the HPA axis. Studies in murine models have shown that caffeine stimulates the production of corticosterone, ACTH, and cortisol [94]. Similar results have been observed in humans [95,96], suggesting that caffeine’s effects are mediated at the CNS level. However, the precise mechanisms underlying caffeine’s impact on the HPA axis remain incompletely defined. Evidence also suggests that caffeine may directly promote cortisol production by acting on adrenal gland cells. While caffeine-induced increases in cortisol can be observed at rest and in a dose-dependent manner—elevating cortisol peaks and prolonging their duration [95]—this response is further amplified when the organism is exposed to additional stressors, such as noise, mental stress, or exercise [96].

Other dietary habits, such as consuming high-carbohydrate meals, have been shown to prolong the cortisol response [97]. This effect is also observed in healthy individuals facing socially stressful events while consuming high-sugar foods, such as glucose or highly sugary juices [98,99]. This phenomenon is primarily attributed to the facilitation of HPA axis reactivity. Nonetheless, further studies are needed to fully elucidate the underlying mechanisms and their implications for overall health.

In summary, dietary habits, including fasting, alcohol consumption, caffeine intake, and high-carbohydrate meals, can significantly impact cortisol levels and the HPA axis. Understanding these effects is crucial for developing dietary recommendations that balance metabolic benefits with the potential risks associated with altered cortisol responses.

#### 3.1.4. Influence of Cancer and Chemotherapy in Cortisol Response 

While cancer itself is not directly considered a disruptor of circadian rhythms, it can significantly influence them. The cancer process represents a constant source of physical and emotional stress for the patient, which is reflected hormonally. Evidence indicates that various types of cancer, including breast cancer, gastrointestinal cancer, leukemia, lung cancer, and prostate cancer, are associated with elevated cortisol levels compared to healthy individuals [100,101,102,103,104,105,106,107,108,109,110,111,112,113,114,115,116]. Furthermore, metastatic cancer patients exhibit higher cortisol levels than those newly diagnosed with breast or prostate cancer [100,114]. In some cases, such as renal cancer, cortisol levels are directly proportional to the degree of malignancy [115]. Additionally, up to 50% of cancer patients experience alterations in sleep patterns that can further exacerbate circadian rhythm disturbances [117]. Lastly, while not universally applicable, various agents used in cancer therapy can lead to cortisol hyperproduction or HPA disruption [118,119,120,121]. 

#### 3.1.5. Implications of Systemic Cortisol Deregulation in Adipogenesis 

Elevated levels of cortisol can circulate throughout the body, affecting various physiological functions. Within adipose tissue—whether subcutaneous or visceral (surrounding internal organs)—several cell types are present, including vascular cells, smooth muscle cells, adipocytes, preadipocytes, and mesenchymal stem cells (MSCs). MSCs are a heterogeneous group of fibroblast-like cells known for their multipotency, which allows them to differentiate into various cell types, such as osteocytes, chondrocytes, myocytes, and adipocytes [122]. The process of MSC differentiation into adipocytes involves intricate molecular events, including the activation of specific signaling pathways and transcription factors. These processes are comprehensively detailed in specialized reviews [123,124,125]. Experimental evidence shows that the differentiation of MSCs into adipocytes (adipogenesis) requires the activation of transcription factors from the peroxisome proliferator-activated receptor gamma (PPARγ) and CCAAT/enhancer-binding protein C/EBP families [126]. Cortisol plays a pivotal role in this process, as it strongly induces PPARγ and several C/EBP subtypes, including α, β, δ, γ, and ε [126], which are essential for the phenotypic changes and metabolic adjustments needed for lipid storage. Indeed, glucocorticoids such as cortisol, along with synthetic analogs like dexamethasone, are a requisite for adipogenic differentiation in vitro [123,125,127,128,129,130,131]. This aligns with observations that patients with HPA axis disorders, such as hypercortisolism, exhibit increased fat accumulation [132,133]. Correspondingly, data from a systematic review involving 693,449 participants showed higher BMI in shift-workers who exhibit higher cortisol levels [134,135,136,137]. 

How, then, does the organism maintain such low differentiation rates—estimated at less than 1% of preadipocytes differentiating during a typical day—in the presence of a daily differentiating stimulus like cortisol? This is particularly intriguing given that ghrelin, prolactin, insulin, and cortisol collectively provide all the essential signals necessary to trigger preadipocyte differentiation on a daily basis. In this context, Bahrami-Nejad and colleagues elegantly demonstrated how the organism establishes a transcriptional circuit that acts as a filter, regulating adipogenesis [138]. Through a series of in vitro and in vivo experiments, they showed that the pro-adipogenic factors PPARγ and C/EBP/β have relatively short half-lives and that a healthy cortisol rhythm—characterized by a robust, singular morning peak followed by a gradual decline throughout the day—is insufficient to drive adipogenesis. This process is not facilitated by merely increasing the stimulus magnitude or shortening the interval between pulses. Remarkably, cells committed to differentiation were found to undergo this process even in response to lower amplitude cortisol stimuli, provided the signal remained consistent over 12 to 18 h. This suggests that certain modern lifestyle habits, as discussed earlier, may modify the cortisol signature in a manner sufficient to activate the transcription factors regulating adipogenesis. These different scenarios are illustrated in Figure 1.

## 4. Impact of Circadian Disruption in the Tumor Microenvironment Remodeling to Promote Cancer Progression 

Previously, we discussed how disruptions in cortisol circadian rhythms can influence adipogenic promotion. In this section, we will explore how elevated cortisol production and adipogenesis remodel the tumor microenvironment (TME) to favor the development, progression, and protection of various types of cancer. 

### 4.1. The Tumor Microenvironment 

The concept of cancer as a single group of altered cells is no longer viable. Cancer is now understood as a complex, dynamic interaction among heterogeneous cells, creating a microenvironment alongside cancer cells. These cancer cells can interact with and influence the behavior of microenvironmental cells to promote their own survival, localization, metabolism, proliferation or even to evade immune responses and drug treatments. This results in the formation of a tumor microenvironment (TME), primarily composed of stromal cells/MSCs, adipocytes, immune cells, as well as non-cellular components such as exosomes, microvesicles, extracellular matrix, and hormones. Adipocytes are a major constituent for multiple cancer microenvironments, such as breast, renal, and ovarian cancer. Although the composition of the TME largely depends on the type and stage of the cancer, the ratio of cellular components may vary. Consulting specialized references for more detailed information is recommended [139,140,141,142]. Circulating cortisol can remodel the microenvironment, affecting both cellular and non-cellular components, which will be discussed below.

### 4.2. Effect of Cortisol on the Cellular Components of the TME 

#### 4.2.1. Adipose TME as Modulator of Chemotherapy

Adipocytes, which constitute adipose tissue, are considered a major component of the organism with active metabolic and endocrine roles. This characteristic can be exploited by certain types of tumors originating from adipocyte-rich sites, such as breast, ovarian, colorectal, pancreatic cancers, and bone marrow leukemias. Physically, an adipose microenvironment creates a protective barrier that impedes the effective delivery of anticancer drugs. Adipocytes can sequester and metabolize various chemotherapeutic agents, such as daunorubicin, mitoxantrone, and vincristine, due to their significantly increased levels of drug-metabolizing enzymes, including aldo-keto reductases and carbonyl reductases [143]. Additionally, studies in murine models have shown that adipose mass can alter drug biodistribution, leading to up to a 28% increase in drug levels in order to obtain the same circulating levels [144]. Interestingly, for certain cancers such as leukemia, adipocytes from multiple sources (subcutaneous, visceral, muscular, pulmonary) secrete chemotactic signals like the cytokine CXCL12, which attract tumor cells. In the context of chemotherapy, this interaction allows tumor cells to protect themselves and evade the effects of daunorubicin or vincristine [145]. Overall, it can be inferred that cortisol-mediated adipocyte accumulation in various tissues may provide a sanctuary for tumor cells by offering physical protection against chemotherapeutic agents, particularly for cancers originating in adipose-rich tissues, such as breast or prostate cancer. Additionally, the accumulation of adipocytes in different deposits (visceral or subcutaneous) may represent secure sites where non-solid tumors, such as leukemias, can disseminate safely throughout the body. However, specific studies are needed to address these observations and confirm these potential mechanisms.

#### 4.2.2. Effect of Cortisol on Immune Cells in the TME

Within the tumor microenvironment, various types of immune cells are present, primarily macrophages and lymphocytes (B, T, and natural killer cells). Dysregulation of cortisol, driven by undesirable modern habits, can readily enter the bloodstream and various compartments, including the tumor microenvironment, disrupting the cellular balance in terms of both number and function. Since their discovery, endogenous glucocorticoids have demonstrated significant immunosuppressive effects, particularly by affecting the cell cycle, arresting lymphocytes, and promoting their death via the mitochondrial apoptotic pathway (reviewed in [146]).

In the context of cancer, persistent cortisol levels can potentially have a negative impact on specialized immune cells with anticancer activity, whether they are residents of the microenvironment or circulating cells patrolling for threats such as cancer (e.g., NK or cytotoxic T cells). The effects of glucocorticoids induced by various stress pathways on NK cells have been shown to depend on their concentration and duration; however, they generally cause marked lymphopenia in lymphocyte subpopulations, including those with specialized anticancer functions (e.g., CD8+ T cells) [147]. Cortisol also reduces the NK’s ability to recognize and form conjugates with target (cancer) cells, as well as decreases the expression of molecules that trigger their activity (e.g., Natural cytotoxicity receptors; NCRs/NKp) or effector molecules (e.g., perforin or granzyme A) [148,149,150,151,152]. 

Macrophage infiltration into the TME is another hallmark of tumor progression [153,154]. Activated macrophages are generally divided into two groups: M1-type macrophages (M1) and M2-type macrophages (M2); both groups are closely associated with inflammatory responses. Proinflammatory M1 macrophages are characterized by increased production of proinflammatory cytokines IL-1, IL-6, and TNFα, in contrast to suppressor M2 macrophages, which produce high levels of IL-10, IL-4, and IL-13 [155]. Clock genes have been shown to be involved in the regulation of M1/M2 macrophage polarization [156]. The M1/M2 ratio is more favorable for the M1 phenotype in the white adipose tissue in obesity [157] and in the early stages of cancer [154]. However, as cancer progresses, the process of tumor-associated macrophage (TAM) polarization comes under the direct control of cancer cells in the TME, and the population of M2-like cells increases dramatically [154]. The M2 phenotype appears to be involved in creating the immunosuppressive microenvironment required for cancer cells to evade host immune control. As shown in various cancer types, both cancer cells and M2 TAM express elevated levels of programmed T cell death ligand 1 (PD-L1), which is well known as an immune checkpoint for cytotoxic CD8+ T cells, inhibiting their ability to kill cancer cells [154,158,159,160]. Analysis of high-throughput sequencing data from thousands of cancer samples has shown that cancer is closely associated with abnormal circadian clocks, as well as global activation of the immune inhibitory molecules PD-L1 and cytotoxic T lymphocyte antigen 4 ligand CTLA-4, anergy and exhaustion of T cells [161].

In summary, at the microenvironmental level, circadian health dysregulation promotes cancer progression by impairing the immune response through the suppression of CD8+ T cells, NK cells, and their effector molecules, as well as disrupting the cytokine profile. 

#### 4.2.3. Adipocyte Recruitment by Tumor Cells

The established relationship between cancer and obesity is well-documented. However, the magnitude of their interdependence continues to be explored. In this context, cancer cells not only seek refuge in adipose microenvironments but also actively generate these niches by recruiting adipose cells to create optimal conditions for their growth. In a study utilizing transgenic mice constitutively expressing GFP, adipose stromal cells (characterized as CD45− CD31− CD29+) were isolated from white adipose tissue. These GFP+ adipose cells were administered intravenously to BALB/c mice that had undergone a mammary allograft. Monitoring the distribution of the adipose cells revealed a preferential localization within tumor sections, with minimal presence in other areas of the organism. These findings were further corroborated in a xenograft model of lung carcinoma. Additionally, tumor size monitoring demonstrated that the adipose cells significantly enhanced the growth of both xenograft and allograft tumors [162]. Similar in vitro findings were observed using adipose tissue-derived cells. The collected supernatant from the adipose cell culture was administered to squamous cell carcinoma cultures, resulting in increased cancer cell proliferation, neovascularization when cocultured with HUVEC cells, and a heightened invasive or migratory phenotype [163]. These results highlight the critical dependence between cancer and adipocytes in malignancy development, particularly by fostering their interaction and promoting paracrine stimulation. The potential soluble factors responsible for these effects will be discussed in the following sections.

### 4.3. Impact of Circadian Disruption on Non-Cellular Elements of the TME

In addition to the cellular modifications induced by systemic cortisol elevations in the TME, significant alterations include modifications in the microenvironment’s components of the extracellular matrix (ECM) and the availability of soluble molecules as signaling cues, as well as metabolic supplies that support cancer development and survival. These modifications are discussed in detail in the following sections.

#### 4.3.1. Extracellular Matrix Remodeling and Tumor Progression

The integrity of the ECM is essential for maintaining tissue homeostasis, as it provides structural support and modulates cellular behavior through interactions with cell surface receptors. This delicate balance is orchestrated by various cells and enzymes that regulate ECM composition and integrity. Disruption of this regulation, often due to aberrant activity of matrix-degrading enzymes or altered cell signaling, can lead to pathological ECM remodeling.

In adipocyte-rich TME, the expression of matrix metalloproteinases (MMPs) such as MMP-11 and PAI-1 is significantly upregulated [164]. This upregulation facilitates ECM remodeling by degrading structural components such as collagen VI, thereby creating a permissive environment for tumor dissemination and invasion, a phenomenon observed across various cancer types [165,166]. Notably, these changes do more than merely facilitate cancer cell invasion; they actively support cancer cell survival. In vitro studies mimicking the TME have shown that breast cancer cells exposed to wild-type MMP-11, but not a catalytically inactive mutant, exhibit prolonged survival, as demonstrated by TUNEL assays and flow cytometry. This survival is limited by batimastat, a broad-spectrum MMP inhibitor, confirming that the catalytic activity of MMP-11 functions both in ECM remodeling and as a direct anti-apoptotic factor in cancer cells. Moreover, high MMP-11 expression has been linked to a reduced immune response, particularly through decreased CD8+ T cell infiltration into the TME [82,167,168]. 

Importantly, multiple MMPs (MMP-2, 3, 7, and 11) are overexpressed in response to elevated cortisol levels exceeding 100 nM, but not at lower concentrations [169]. Similarly, PAI-1 is directly regulated by glucocorticoids and cancer-associated adipocytes [164,170], contributing to the promotion of proliferative signals, evasion of tumor cell death, angiogenesis, and the invasion or persistence of certain cancer types [171,172]. 

Other factors influencing the extracellular matrix include collagen beta-galactosyltransferase 2 (COLGALT2), an enzyme responsible for collagen glycosylation. This process enhances collagen stability and promotes cellular adhesion. However, studies have shown that adipocytes derived from mesenchymal cells can drive the invasion, migration, and proliferation of osteosarcoma cells through the overexpression of COLGALT2. This overexpression leads to increased extracellular matrix remodeling and creates a microenvironment that supports cancer development [173].

#### 4.3.2. Secretome-Mediated Modulation of the TME

Dysregulation of systemic cortisol not only induces cellular changes but also impacts soluble molecules within the microenvironment, including proteins, microRNAs, lipids, growth factors, cytokines, chemokines, hormones, and enzymes. For example, shift workers with elevated cortisol levels exhibit altered cytokine profiles in unstimulated PBMCs, characterized by increased IL-6 and decreased TNF-alpha, alongside suppressed cytokine responses to activating stimuli [174,175,176]. Similarly, adipocytes in the TME are influenced by cancer cells to produce elevated levels of IL-6, which enhances malignancy by promoting invasiveness and other hallmarks of cancer [164,177]. Conversely, the effects of reduced TNF-alpha on cancer are not straightforward, as they vary by cancer type. However, decreased TNF-alpha limits NK cell infiltration into the TME, reduces their cytotoxic potential, and disrupts the balance between activating and inhibitory molecules, thereby indirectly facilitating cancer progression.

Additionally, sleep disorders are associated with alterations in circulating microRNA profiles, such as miR-188, which promotes the differentiation of bone marrow MSCs into adipocytes. This creates favorable niches for the colonization and protection of cancer cells that prefer these microenvironments, such as leukemias [65,66]. In an obese mouse model, the increase in adipocytes at the subcutaneous, perirenal, epididymal, and bone marrow levels supports the incorporation of cancer cells. This phenomenon is attributed to MSCs, pre-adipocytes, and adipose tissue producing and secreting the chemokine CXCL12, which not only facilitates the homing of cancer cells but also, through its receptors CXCR4 and CXCR7, influences gene expression and proliferation in various cancers [145,178]. Similarly, the adipocytes from periprostatic tissue produce large amounts of the CCL7 chemokine to favor prostate cancer cell migration [179].

The exosomes derived from adipocytes have been shown to secrete factors that significantly alter the phosphorylation states of cancer cell proteins, thereby promoting cell cycle continuity and proliferation in cancers highly dependent on the adiposity of the microenvironment, including both ER-positive and negative breast cancer [180]. 

Cortisol-induced adipogenesis leads to the increased production of leptin primarily by adipocytes [181]. Consequently, leptin acts on its receptors located on various cancer cells, which stimulates the overproduction of MMPs and subsequently enhances cancer cell invasion and progression [92,182]. In addition to its role in remodeling the TME and promoting cancer invasiveness, leptin functions as a growth factor for numerous cancer types. However, the impact of leptin varies depending on the specific cancer type [183,184,185].

Finally, persistent cortisol induces resident stromal cells within the TME to produce and release significantly higher amounts of estrogens—ranging from 5 to 7 times more—through the activation of the enzyme aromatase. This process creates an environment that promotes the progression of estrogen-dependent cancers [126,186].

#### 4.3.3. Metabolic Trading in the TME

Within the TME, the persistent impact of systemic cortisol is reflected in elevated glucose levels, as glucocorticoids reduce glucose uptake and oxidation in skeletal muscle [187]. Additionally, the adipocytes generated due to systemic cortisol dysregulation can actively influence cancer cell metabolism at multiple levels. A hallmark of cancer cells is their high metabolic plasticity, which refers to their ability to adapt metabolism in response to various microenvironmental and physiological conditions. One well-known adaptation is the Warburg effect, where cancer cells preferentially engage in aerobic glycolysis, producing lactate instead of fully oxidizing glucose to generate energy through oxidative phosphorylation in the mitochondria. In this context, prostate cancer cells have been shown to undergo metabolic reprogramming in the presence of adipocytes, increasing the expression of glycolytic enzymes, which enhances lactate production and reduces mitochondrial activity [188]. This reprogramming allows the cancer cells to meet their rapid growth and proliferation demands by utilizing glucose to produce metabolic intermediates necessary for biomolecule synthesis. Furthermore, lactate production contributes to the creation of acidic microenvironments that promote tumor invasion and suppress immune responses.

By employing glucose for these purposes, cancer cells require additional energy sources and, therefore, promote adaptations, such as the utilization of free fatty acids (FFAs); thus, they activate the adipocytes to produce and release FFAs resulting from lipolysis, which are subsequently used via beta-oxidation. This process optimizes cancer metabolism and proliferation, as observed in breast, prostate, and ovarian cancers and leukemias [179,188,189]. Additionally, elevated circulating cortisol levels have a direct effect on systemic adipose tissue, promoting lipolysis and the release of fatty acids [190,191,192], creating a favorable environment for cancer development, as seen in patients with lung, gastric, thyroid, rectal, colon, and ovarian cancers [193].

Furthermore, the metabolism of some cancers may rely on amino acids. Certain cancers have been described as being “addicted” to amino acids such as glutamine or asparagine [194,195,196]. In this context, adipocytes in the bone marrow microenvironment of patients with acute lymphoblastic leukemia (ALL) exhibit elevated levels of the enzymes necessary for glutamine synthesis [197]. Correspondingly, in a syngeneic murine leukemia model, the efficacy of asparaginase, a key enzyme in leukemia treatment that hydrolyzes asparagine and glutamine, was impaired in the presence of adipose-rich TME, which provides aa to the ALL, thus promoting leukemia development. These data suggest that adipocytes collaborate with cancer cells to protect and facilitate their adaptation by optimizing glucose utilization for biosynthetic blocks, using free fatty acids, and exploiting amino acids derived from adipocytes. Although these adaptations may be shared among various cancer types, the impact of the adipose secretome may depend on the malignancy state or specific characteristics of each cancer subtype [198].

In summary, the data presented here demonstrate that cortisol dysregulation and the promotion of adipogenesis significantly influence the tumor microenvironment, enhancing and protecting cancer development. However, this perspective may be considered reductionist, as the influence of an adipose TME in cancer progression needs to be studied extensively for each cancer type. The content of this chapter is summarized in Figure 2.

## 5. Melatonin in the Crosstalk of Circadian Disorder, Obesity and Cancer

### 5.1. Melatonin Is a Pleiotropic Physiological Regulator

Melatonin is an indolamine (*N*-acetyl-5-methoxytryptamine) produced by all living organisms, including microorganisms, plants and animals. Several detailed reviews on melatonin synthesis and its physiological roles are recommended [199,200,201,202]. Here, we focus on the relation of melatonin disorders to cancer and obesity risks.

Melatonin is suggested to have appeared originally in primitive bacteria and acted as an antioxidant [201,202]. According to the endosymbiotic theory of the origin of eukaryotic cells, the proteobacteria and cyanobacteria were ingested and evolved into mitochondria and chloroplasts, correspondingly. Later on, eukaryotic cells conserved the capacity to produce melatonin in mitochondria. Accordingly, all plant and animal cells produce melatonin, which primarily acts as a scavenger of reactive oxygen and nitrogen species, ROS, and RNS. Primary melatonin functions as ROS, and RNS scavenger is independent of any receptors. Important melatonin functions as a key regulator of the sleep–wake cycle and internal circadian rhythm that appeared later in animal evolution and is related to the pineal gland. The HPA axis, following a circadian rhythm, oppositely regulates cortisol and melatonin synthesis, balancing each other [203]. 

Two plasma membrane G protein-coupled melatonin receptors, MT1 and MT2, have been described [200,204]. However, melatonin can also act via binding to nuclear RZR/ROR and cytosolic MT3 (quinoline reductase QR2) receptors, as well as to some other receptors (5-HT2C receptor, β2-adrenoceptor, A2A adenosine receptor, GPR50), channels (Cav2.2 channels), and transporters (GLUT1) [200,205]. Importantly, melatonin receptors and target molecules are widely distributed in the central nervous system and peripheral tissues, explaining a very broad spectrum of its physiological effects. Melatonin acts as an antioxidant, regulator of circadian rhythms and sleep, an immunomodulator, and an anti-inflammatory and anticancer agent [200,201,206]. The capacity of melatonin to promote the formation of tunneling nanotubes for intercellular mitochondria transfer and a reshape of the mitochondrial network in some cells after injury was demonstrated recently [207,208,209]. Melatonin effects can be classified as (1) immediate effects, such as antioxidant effect; (2) prospective effects, triggered by melatonin in the night but observed during the next day; (3) chronobiotic effects, such as regulation of clock genes in peripheral tissues, caused by repeated daily melatonin signals; (4) seasonal effects in regulation of reproduction, energy metabolism, immune responses, and body weight control, determined by specific melatonin profile in every individual; (5) transgenerational effects of maternal melatonin crossing the placenta [200,206]. 

To date, we have been unable to determine a relative distribution of pineal and extra-pineal melatonin across different tissues. It appears that melatonin is present in the bloodstream, acting at the systemic level, and primarily originates from the pineal gland. In peripheral tissues, however, melatonin seems to be produced independently of light/dark cycles. While melatonin concentrations in mammalian blood can reach nanomolar levels, extra-pineal concentrations vary significantly across tissues and can reach micromolar levels. Notably, exogenously administered melatonin tends to concentrate in mitochondria. [210].

### 5.2. Factors Affecting Melatonin Production in Humans

The environmental light–dark cycle is widely recognized as the primary regulator of pineal melatonin synthesis, with significantly higher melatonin levels observed during the night and in darkness [199,211]. Exposure to light, particularly in the blue spectrum, activates the retinal system and inhibits the release of sympathetic noradrenaline from the pineal gland, effectively suppressing melatonin production [212]. A recent study in healthy volunteers demonstrated a rapid suppression of melatonin synthesis upon nocturnal light exposure, with corresponding increases in cortisol levels that varied significantly based on light characteristics and exposure conditions [74].

Melatonin production varies widely between individuals, with nocturnal serum levels ranging from 0–20 pg/mL to 50–200 pg/mL in humans [213]. Age is one of the most extensively studied factors influencing this variability. Melatonin synthesis begins to follow a circadian rhythm in infancy, emerging around 5–6 months of age. Peak nocturnal levels occur between 5–10 years old, followed by a progressive decline throughout life [213,214]. While most elderly individuals experience a significant reduction in melatonin production, some aged 70–90 continue to exhibit relatively high levels [215]. This decline is often linked to structural changes in the pineal gland, such as calcification and increased glial tissue volume [216].

Age also significantly influences the effect of ALAN on melatonin suppression, with younger individuals showing greater sensitivity. For example, children experience nearly double the suppression rates of melatonin under ALAN compared to adults, with suppression rates of 88.2% in children versus 46.3% in adults under moderately bright light [216,217]. In animal models, aging-related changes in the SCN further support this differential sensitivity. Older mice display alterations in GABAergic function and calcium homeostasis, which disrupt the excitatory–inhibitory balance within the SCN, reducing its ability to maintain circadian synchrony [218]. Additionally, the SCN of aged mice exhibits a marked reduction in the number and area of GABAergic synaptic terminals, suggesting that aging impairs the SCN’s ability to regulate circadian rhythms effectively [219]. These findings indicate that both structural and functional changes in the SCN with age contribute to the altered sensitivity to ALAN and the overall decline in melatonin production in older individuals.

There is considerable individual variability in melatonin production, with nocturnal serum melatonin levels ranging from 0–20 pg/mL to 50–200 pg/mL in humans [213]. Age is one of the most extensively studied factors affecting melatonin production. It is well established that melatonin synthesis begins to follow a circadian rhythm in infancy, emerging at around 5–6 months of age. Peak nocturnal melatonin levels occur between the ages of 5–10 years, after which levels decline progressively throughout life [213,214]. While a significant decrease in melatonin production is observed in the majority of elderly individuals, some individuals aged 70–90 years continue to exhibit relatively high melatonin levels [215]. Gender differences in melatonin production have been noted in older people, with women displaying higher levels than men [215]. The age-related decline in melatonin production is thought to result from degenerative changes in the pineal gland, including calcification and an increase in glial tissue volume [216].

Melatonin is synthesized from its precursor, the essential dietary amino acid L-tryptophan, with several vitamins and minerals (including folate, vitamin B6, zinc, and magnesium) serving as cofactors. Some studies have explored the impact of diet on the efficiency of melatonin synthesis [220]. Severe dietary restrictions may impair melatonin production, among other physiological effects. For instance, reduced plasma tryptophan levels have been observed in depressed patients [221]. In experimental settings, acute dietary tryptophan restriction led to a reduction in nocturnal melatonin secretion in both male and female volunteers [222]. However, severe dietary deficiencies are relatively rare, and fasting has been shown to have no effect on nocturnal melatonin production, with minimal impact on daytime levels [220].

### 5.3. Melatonin Possesses Anticancer Properties 

Melatonin tends to oppose tumorigenesis [213,223,224,225,226,227]. Due to its antioxidant properties, melatonin can prevent DNA damage, contribute to DNA reparation, and maintain the genomic integrity of cells, which finally prevents a malignant transformation [224]. Further, melatonin can regulate cell proliferation, DNA synthesis, and cell survival vs. cell death signaling [224,225,226]. 

Melatonin inhibits the proliferation of numerous cancer cell lines, including breast, prostate, lung, and bladder cancers, melanoma, and glioblastoma, among others [224,226]. This direct anticancer effect is related to the melatonin interaction with the components of various signaling pathways involved in tumorigenesis, such as mTOR/Akt, Wnt, JNK/MAPK, and regulatory miRNAs, among others. Melatonin binding to specific melatonin receptors seems to be important for its anticancer effects, at least in some cancer types. For example, in breast cancer, melatonin was shown to inhibit tumor growth and metastasis through the interaction with MT1 receptors [223]. The anticancer effect of melatonin in some types of cancer was shown to be related to its capacity to influence DNA methylation [228]. In Warburg-dependent cancers, melatonin can reverse Warburg-type metabolism, causing metabolic collapse and cell death [209,229,230,231]. These properties are likely attributable to melatonin’s role in regulating glucose metabolism, particularly by inhibiting glycolysis and mitochondrial respiration. As previously mentioned, melatonin acts through C/EBP to inhibit PPARγ, which is a key factor for many glycolytic genes, including hexokinase, phosphofructokinase, and pyruvate kinase [200,232,233]. 

Additionally, melatonin has the ability to inhibit multiple transporters responsible for multidrug resistance, such as adenosine triphosphate-binding cassette (ABC) transporters, in various types of cancer [234,235,236,237]. Consequently, the effects of melatonin on these proteins may enhance the retention of chemotherapy agents used and limit cancer progression.

According to clinical observations reported by different independent research groups, nocturnal melatonin concentration levels are significantly decreased in patients diagnosed with different cancers, including breast, endometrial, lung, stomach, colorectal, ovarian, and prostate, as compared to healthy individuals (reviewed in [214]). In this relation, breast and endometrial cancer risk was shown to be increased in postmenopausal shift-working women [238]. Night shift work represents a risk factor for endometrial cancer, in particular in obese women [238]. A significant negative correlation between the level of plasma melatonin and endometrial cancer in humans has been reported [239]. 

Results of several clinical trials using melatonin as an anticancer adjuvant therapy, with daily doses in the range of 3–20 mg, were reported [224]. Among the observed effects were improved sleep and health-related life qualities, a decreased risk of developing depressive symptoms, and a better score in social well-being. However, directed anticancer effects related to tumor growth and survival rate were not reported in these trials. Some recent work proposed melatonin-loaded nanocarriers to improve their specific targeting and therapeutic effects [240]. 

Melatonin synthesis was shown to be suppressed by ALAN-induced circadian disruption and circadian oscillations in animal models where rats were exposed to dim ALAN light during night [241]. For these animals, in contrast to animals with a normal 12 h/12 h light/darkness cycle, an increased growth of human breast cancer xenografts was observed. Important metabolic disorders were also reported, such as 24-h hyperinsulinemia and hyperglycemia with elevated blood IGF-1 levels, which points to a relationship between circadian and metabolic disruptions and cancer growth. 

### 5.4. Melatonin Can Prevent the Formation of Adipose Microenvironment

Melatonin is now recognized as a potential regulator of metabolism (reviewed in [206]). It regulates the synthesis of GLUT1 and leptin, insulin secretion, and glucose metabolism and induces nocturnal insulin resistance and diurnal insulin sensitivity in accordance with nocturnal fasting and diurnal feeding. Besides, melatonin is an important modulator of AT deposition as it participates in the regulation of lipogenesis and lipolysis, synthesis and breakdown of glycogen in the liver and muscles, recruitment of BAT, and browning of WAT. 

Mechanistically, melatonin is capable of inhibiting essential transcription factors for adipogenesis, such as C/EBPβ (CCAAT/enhancer-binding protein beta) and PPARγ (peroxisome proliferator-activated receptor gamma), as discussed in Section 3.1.5 [242].

Laboratory evidence supports a relationship between decreased melatonin production and obesity [243,244]. Given that adipose tissue serves as a protective niche for chemoresistant cancer cells, including cancer stem cells, in cancer patients, melatonin may indirectly reduce cancer development by diminishing the formation of the adipose TME [245]. As was reported in animal models, supplementation of external melatonin caused a reduction in weight gain and a decrease in visceral fat deposition [246,247]. 

Here, we present reported clinical evidence on how melatonin supplementation affects some physiological parameters, including weight and AT accumulation in different pathophysiological conditions in humans. Decreased melatonin production and body weight gain were shown to be associated with menopause [248,249,250,251]. In post-menopausal women, compensatory adjuvant treatment with low doses of exogenous melatonin (1–3 mg daily, for one year) demonstrated a beneficial effect on body composition, with a reduced fat mass and increased lean mass, without affecting body mass index [252]. These data were in accordance with the increase of plasma adiponectin in melatonin groups, with a higher level at a higher melatonin dose (3 mg).

Prolonged melatonin administration at higher doses (5–8 mg daily for 6–12 months) was reported to decrease BMI in postmenopausal obese women and improve psychosomatic symptoms and sleep quality [253,254]. 

Shift work at nighttime was shown to cause circadian misalignment, which is associated with reduced production of melatonin, sleep disorder, and weight gain and can contribute to the increased risk of various medical conditions [255]. In this regard, the effect of working night shifts is different in different chronotypes, with early chronotypes showing a more pronounced circadian misalignment as compared to late ones [255]. Although there were various publications regarding the effect of exogenous melatonin on shift work sleep disorder [256], it was the only study about melatonin’s effect on body weight gain in shift workers [257]. In this study, melatonin administration (3 mg daily before sleeping for 24 weeks) has demonstrated a benefic effect in the reduction of circadian disturbance. Additionally, weight loss reducing BMI, hip, and waist circumference were observed in the group of overweight nurses of early chronotype, in comparison with the placebo group [257]. 

Several pieces of evidence regarding exogenous melatonin effects on weight gain come from clinical reports with patients receiving second-generation antipsychotics (SGA) to treat bipolar disorder or schizophrenia. In these studies, melatonin as adjuvant therapy was expected to prevent metabolic side effects of SGA, which cause an increase in triglyceride and cholesterol levels and weight gain [258]. In a randomized, double-blind, placebo-controlled study in patients with a first episode of schizophrenia, reduced metabolic side effects of olanzapine were observed when melatonin was co-administrated [259]. In particular, abdominal obesity, hypertriglyceridemia, and weight gain were attenuated. Interestingly, a similar protective effect of melatonin was observed in patients with bipolar disorders treated with different SGA but not with schizophrenic patients [260]. Mostafavi and colleagues have undertaken a meta-analysis of seven clinical trials with a total of 244 patients, where melatonin was used as an adjunctive treatment to medication, causing an increase or decrease in a patient’s body weight [261]. They concluded that melatonin possesses a buffering effect on body weight fluctuations: when standard treatment caused body weight changes, increase or decrease, melatonin diminished both of these effects. The buffering effect of melatonin was more pronounced in children and adolescent patients.

Melatonin was suggested as an adjuvant therapy for the treatment of patients with non-alcoholic fatty acid disease (NAFLD) [262]. In parallel with the improvement of many factors related to NAFLD, significant reductions in weight, waist, and abdominal circumference were observed in patients receiving melatonin (6 mg daily, 12 weeks) compared with the control group.

Melatonin replacement therapy (3 mg daily) was proposed for four patients with surgical removal or radiotherapy of the pineal gland due to pineal tumors. Although body weight was not changed within the observation period, the volume and activity of BAT were increased, as evaluated by positron emission tomography MRI [263]. 

In summary, melatonin plays a crucial role in regulating metabolism and cancer development, particularly through its influence on adipogenesis and immune modulation. Its production is significantly affected by circadian disruptions, increasing cancer risk in vulnerable populations. Future research should focus on elucidating the molecular mechanisms linking melatonin to cancer pathways to harness its therapeutic potential in prevention and treatment.

## 6. Perspectives

The intricate relationship between circadian rhythms, hormonal regulation, and the TME highlights the critical role of cortisol in cancer development and progression. While it is well understood that cortisol promotes changes favorable to cancer, such as the creation of adipose niches that protect cancer cells, it is also essential to explore other aspects. Notably, various types of cancer exhibit circadian behavior, express and depend on the clock genes, and, therefore, are considered peripheral clocks (Appendix A). Furthermore, some of these clock genes are regulated by cortisol levels, as they contain glucocorticoid response elements. This regulation adds another layer of complexity to the circadian control of cancer progression.

This interaction holds significant clinical relevance because the expression of clock genes and their proteins, such as CLOCK and BMAL1, has been shown in certain cancers, including leukemia, to acetylate glucocorticoid receptors. This acetylation inhibits the receptor’s function, promoting resistance to glucocorticoid-based therapies [264]. As a result, understanding the circadian regulation of cancer and its interaction with cortisol may provide insights into new therapeutic strategies aimed at overcoming drug resistance.

The potential of GLP-1 receptor agonists warrants further exploration within the context of circadian health and obesity. These agents exhibit anti-adipogenic effects by suppressing appetite, making them valuable in obesity management. Although some studies suggest that GLP-1R agonists may influence the stress response by activating the HPA axis, the evidence remains inconclusive. For instance, no significant differences in cortisol production have been found between patients treated with dulaglutide and those receiving a placebo, indicating that these agents may not directly impact cortisol levels [265]. However, liraglutide has demonstrated anxiolytic effects in animal models, disrupting anxiety-related behaviors and limiting hedonic food consumption linked to stress, potentially mitigating adipose tissue accumulation [266]. Additionally, liraglutide administration is associated with increased sleep duration in rats, which may help alleviate sleep disturbances [267]. In neurodegenerative disease models, such as Alzheimer’s, GLP-1R agonists have been shown to restore circadian health [268]. Given that GLP-1 production is closely tied to food intake, disruptions in circadian rhythms—such as prolonged fasting or irregular eating patterns—could adversely affect GLP-1 secretion and its metabolic roles. Therefore, it is crucial to investigate how different circadian disruptors influence GLP-1R expression and function to fully understand the implications of these receptors in circadian health.

As discussed, serum melatonin levels are primarily affected in individuals experiencing circadian alterations, such as older people and night workers, and to a lesser extent in those on specific dietary restrictions. This decreased melatonin production can disrupt numerous physiological processes, including glucose metabolism and adipose tissue regulation, which are vital for maintaining metabolic homeostasis. Melatonin not only serves as a metabolic regulator but also exhibits immunomodulatory effects that may influence cancer dynamics [269]. It contributes to cancer control through direct mechanisms, including the inhibition of cancer cell proliferation, and indirect mechanisms by limiting adipogenesis and favoring immune system activity [224,270,271].

Therefore, circadian disruption resulting in elevated circulating cortisol levels and decreased melatonin levels may increase the risk of developing and promoting various types of cancer, particularly in elderly individuals, night workers, and those with obesity or during periods of stress.

While melatonin plays a critical role in circadian regulation in both animals and humans, the evidence supporting its function in humans is comparatively limited. Notably, although melatonin is essential for transmitting information about circadian cycles in both groups, its effects may be more pronounced and direct in animals. This difference may arise from the fact that circadian synchronization in humans is more strongly influenced by social cues than in comparison to animals.

However, emerging evidence suggests that melatonin may exert additional anticancer effects through multiple mechanisms, including its antioxidant, immunomodulatory, and anti-inflammatory properties. Moreover, melatonin may indirectly reduce cancer risk by limiting adipose tissue accumulation and obesity, factors closely associated with the development of certain cancers.

Melatonin supplementation emerges as a promising strategy to mitigate these risks, with clinical evidence showing beneficial effects on body composition and metabolic parameters. Innovative nanoformulations of melatonin hold the potential to enhance its bioavailability and efficacy, positioning it as a strong candidate for therapeutic interventions targeting both obesity and cancer. As research continues to unveil melatonin’s multifaceted roles and interactions with metabolic pathways, completing and promoting clinical trials in humans becomes crucial to strengthening its role as an anticancer agent and furthering its therapeutic applications.

While this study has focused on demonstrating how the dysregulation of cortisol and melatonin creates an adipogenic environment conducive to cancer progression, it is important to note that all the circadian cycle disruptors discussed can potentially modify the response to various medications in terms of absorption, metabolism, distribution, and elimination. This topic is central to chronopharmacology and warrants further investigation, particularly with a focus on understanding its impact on anticancer therapies. Understanding how circadian disruptions affect drug responses could lead to more effective treatment strategies tailored to individual patient biological rhythms.

## 7. Conclusions

Our work highlights the profound impact that modern lifestyles have on the disruption of circadian health, particularly through the imbalance of cortisol and melatonin levels, which can promote cancer development. Such deregulations are not isolated events but rather interconnected; for example, sleep deprivation commonly observed in shift workers is often associated with compounding factors such as operational tasks. These tasks not only limit melatonin production but also induce modifications in cortisol levels. Elevated cortisol regulates feeding behavior—particularly hedonic eating—contributes to insulin resistance and promotes fat accumulation. Collectively, these factors disrupt overall metabolism, contributing to disorders such as obesity and diabetes and, consequently, an oncological event.

While the interconnection of these events is presented in a rational, logical, and coherent manner, larger studies, especially in humans, are needed to strengthen these findings. Furthermore, although neither cortisol nor melatonin directly causes cancer or obesity, it is essential to emphasize that the dysregulation of these components chronobiology significantly impacts the body, creating an adipogenic environment that can foster malignancy.

Many of the disruptors, such as prolonged exposure to electronic devices like cell phones, screens, and external light sources, are difficult to control. Similarly, shift work, which is integral to professions like healthcare (e.g., doctors and nurses), is unlikely to disappear. However, it is essential for the general population to focus on improving modifiable aspects of our lifestyle, such as diet and sleep quality, while paying particular attention to high-risk groups like shift workers and older people.

Therapeutic strategies, including melatonin supplementation, diet adjustments, and regular exercise, can play a crucial role in mitigating these risks by reducing adipose tissue accumulation and lowering the likelihood of cancer protection and drug resistance. These lifestyle modifications, alongside targeted interventions, offer a promising approach to preserving circadian health and minimizing the adverse effects of modern living.

## Figures and Tables

**Figure 1 cancers-16-03706-f001:**
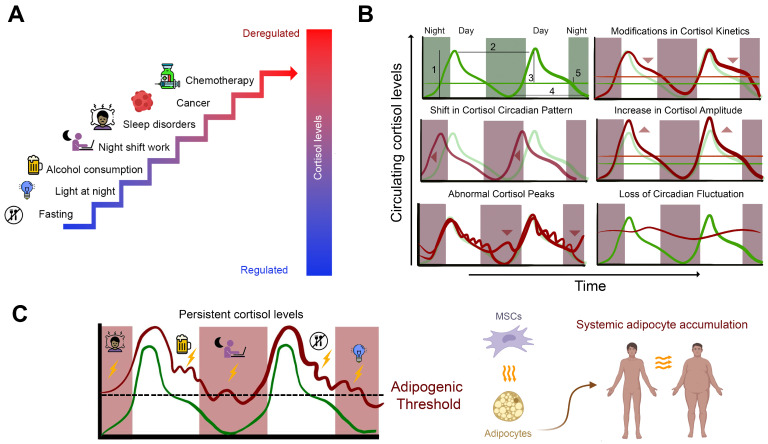
Impact of circadian rhythm disruption on systemic cortisol levels and MSCs differentiation into adipocytes. (**A**) Modern habits disrupting circadian rhythms (discussed in Chapter 3), affecting cortisol production, along with carcinogenic processes and chemotherapy, which also contribute to cortisol dysregulation. (**B**) The upper left panel illustrates healthy cortisol behavior. Chronobiological parameters include (1) the cortisol peak, known as the Cortisol Awakening Response (CAR); (2) the period, the time between peaks; (3) the amplitude, the change from the mesor to the peak; (4) the nadir, the lowest level during fluctuations; (5) the average cortisol level across the day, the mesor (green horizontal line). Red panels and lines demonstrate various modifications to the cortisol signature caused by disruptors, including altered kinetics, shifted CAR timing, changes in CAR or mesor levels, irregular peaks, or loss of period. The graphs are illustrative and based on clinical data from independent studies (Appendix A). (**C**) Accumulation of both subcutaneous and visceral fat results from multiple disruptive events or activities impacting cortisol levels, promoting the adipogenic differentiation of MSCs.

**Figure 2 cancers-16-03706-f002:**
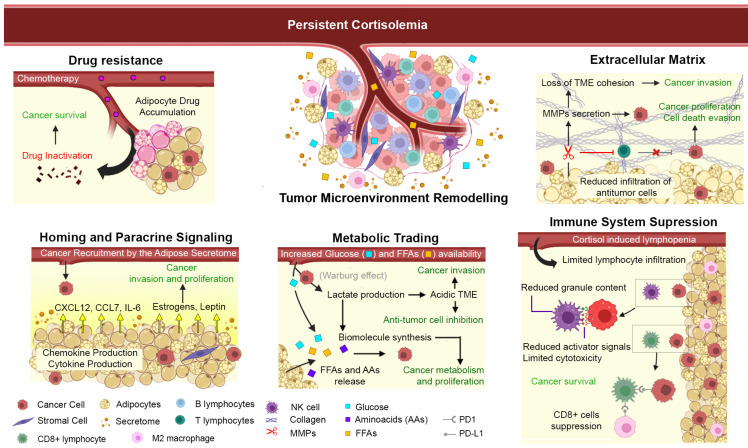
Effects of cortisol dysregulation on the TME and cancer progression. Persistent cortisol elevation induces differentiation of adipocyte precursors across various tissues, leading to adipose enrichment in the TME, which creates conditions that protect different cancer types. Each panel illustrates the protective effects of the adipose-rich TME, including drug evasion, chemotherapy resistance, secretion of survival and proliferation signals, metabolic support, immune suppression, and TME remodeling to facilitate cancer cell invasion and survival. Detailed data are provided in Chapter 4. The bottom section includes the legend for cell types and substances depicted in the panels.

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
