# Peer review of "Impact of Modern Lifestyle on Circadian Health and Its Contribution to Adipogenesis and Cancer Risk"

_cancers, 2024, doi:10.3390/cancers16213706_

Round 1

Reviewer 1 Report

Comments and Suggestions for Authors

This review paper highlights the role of changing lifestyles, and its impact on human health in the perspective of increasing cancer risk and adipogenesis. It describes how disturbance in organisms natural circadian clock by environmental factors disrupts cellular mechanisms that is the cause of generation of diseases such as cancer. This article also reviews the role of melatonin in reducing the incidence of cancer through its role in preventing adipogenesis. Overall, the article is well written in the aspect of lifestyle factors influencing cancer development and the role of melatonin in controlling the rate of tumor formation. However, some discussion points lack specific details or appropriate citations. The authors need to address these issues before publication in Cancers.

Author Response

Reviewer 1

Reviewer comment

Authors response

This review paper highlights the role of changing lifestyles, and its impact on human health in the perspective of increasing cancer risk and adipogenesis. It describes how disturbance in organisms natural circadian clock by environmental factors disrupts cellular mechanisms that is the cause of generation of diseases such as cancer. This article also reviews the role of melatonin in reducing the incidence of cancer through its role in preventing adipogenesis. Overall, the article is well written in the aspect of lifestyle factors influencing cancer development and the role of melatonin in controlling the rate of tumor formation

Thank you for your positive feedback on the structure and writing of the manuscript

some discussion points lack specific details or appropriate citations. The authors need to address these issues before publication in Cancers.

Accepted and modified:

Thank you for your feedback. In the current version, you will find a much more detailed manuscript and a discussion enriched with appropriate citations to enhance audience appreciation.

Introduction. Page 3, line 93. Expand NMDA

Accepted and modified: The meaning of NMDA acronym was stated. We appreciate your guidance in improving the manuscript

Introduction pg. 3, line 95. Mention the names of the signalling pathways activated.

Accepted and modified: The pathways were defined, and corresponding references were added.

Page 3, line 120. Please clarify what is meant by humoral factors.

Accepted and modified:

We sincerely thank the reviewer for pointing out this important clarification, as it allowed us to correct an oversight present since the first draft. While the suprachiasmatic nucleus can indeed be modulated by humoral factors produced by immune cells, such as cytokines—including TNF-α (https://doi.org/10.1016/j.cytogfr.2019.04.001)—in this section, we were specifically referring to the SCN's ability to regulate other functions through the oscillatory production of TGF-α, not TNF-α. This has been clarified in the revised version, and the necessary modifications have been made and highlighted in the text.

Page 2,3. The referenced cited are extremely backdated. It is recommended to cite recent articles of the field. There are several recent articles published in the field of circadian biology and role of melatonin.

Accepted and explained:

We appreciate the comment regarding the cited references. It is important to note that the content of Chapter 2 specifically focuses on the fundamentals of circadian rhythms and the function of the suprachiasmatic nucleus. In this context, we consider it essential to cite original works and give appropriate credit to the findings achieved with significant technical and methodological complexity. The discoveries generated from these original works remain relevant. Therefore, we believe it is legitimate to retain these references in the manuscript.

Additionally, at the end of this chapter, we have included a couple sentences inviting readers to consult the existing recent articles on the subject that are available.

Page 5, line 239-242. The author may briefly describe how changes in LAN, affects melatonin production.

Accepted and modified:

We appreciate the suggestion and believe that these changes are appropriate; they have been incorporated into the new version of the manuscript, along with the relevant references.

Page 8, line 350. The author should explain full abbreviation when mentioning first time. PPARγ and C/EBP families.

Accepted and modified:

Abbreviations were stated  as recommended.

Page 17, line 756-758. Is the effect of LAN on melatonin in animal models, age dependent or irrespective of the age of the organism? If yes, explain how LAN affects melatonin production in young and old animals.

Accepted and modified:

We appreciate this suggestion and have made the appropriate changes in the section 5.2 (L702) to clarify this point. We believe that this recommendation enhances the manuscript's version and provides a clearer understanding of the effect of light-at-night (LAN) on melatonin production in different age groups.

Author can briefly describe how a disrupted circadian rhythm due to sleep deprivation condition, melatonin production and subsequent cortisol production is connected and its effect on body metabolism.

Accepted and modified:

We appreciate the suggestion and believe it is important to address this point as recommended. Emphasis has been placed in the conclusions section on how these events are not isolated but rather interrelated, contributing to metabolic dysregulation and the promotion of oncological events.

Reviewer 2 Report

Comments and Suggestions for Authors

After careful consideration, I fell that the manuscript entitled Impact of modern lifestyle on circadian health and its contribution to adipogenesis and cancer risk has merit, but is not suitable for publication as it currently stands.

Some of specific problems are listed below.

1. The authors state that the paper reviews scientific literature from Scopus, Google Scholar and Pubmed. However, the selection criteria used to select the studies used in this study are not described. In fact, even though this is not a systematic review, the criteria for selecting the studies should be described in order to avoid bias, since only studies that agree with a certain statement may be selected, while other studies that present divergent results may be neglected. So I think it would be interesting for the authors to include a section presenting the strategies and criteria adopted to search for these articles in these databases (Describing the search strategies, inclusion and exclusion criteria for example).

 2.  Are the figures presented in the manuscript your own? If not, include the sources.

3. A careful review of the all references is necessary. Note, for example, that the reference "Nakamura et al " (line 276) presents the wrong year and the reference on line 926 is not numbered.

Author Response

Reviewer 2

Reviewer comment

Authors response

After careful consideration, I fell that the manuscript entitled “Impact of modern lifestyle on circadian health and its contribution to adipogenesis and cancer risk” has merit, but is not suitable for publication as it currently stands.

Thank you for your careful review and the time you dedicated to evaluating our manuscript, as well as for the positive recognition of its merit. We have addressed the concerns you raised, and you will find that this version is optimized and improved thanks to your valuable feedback.

The authors state that the paper reviews scientific literature from Scopus, Google Scholar and Pubmed. However, the selection criteria used to select the studies used in this study are not described. In fact, even though this is not a systematic review, the criteria for selecting the studies should be described in order to avoid bias, since only studies that agree with a certain statement may be selected, while other studies that present divergent results may be neglected. So I think it would be interesting for the authors to include a section presenting the strategies and criteria adopted to search for these articles in these databases (Describing the search strategies, inclusion and exclusion criteria for example).

accepted and modified:

Thank you for your suggestion as it certainly helped us to improve the current version of the manuscript.

As you correctly noted, our review is not systematic but rather narrative in nature. Its aim is to provide a novel and timely interpretation of the relationship between circadian disruption and cancer propensity, a topic that remains not fully understood. We have dedicated an appropriate paragraph in the introduction to outline our selection strategy for the articles, as well as the inclusion and exclusion criteria, to address your concerns regarding potential bias in the selection of studies. Thank you once again for your constructive feedback.

Supplementary Tables 1 and 2 were not made available, so it was not possible to perform a proper review of the statements made from them.

Explained: We would like to clarify that the supplementary material was provided at the time of manuscript submission and remains available on the platform (please see the attached image). We regret any inconvenience this may have caused and invite you to consult the available material for your favorable reconsideration.

Are the figures presented in the manuscript your own? If not, include the sources.

Explained:

Thank you for your inquiry regarding the figures presented in the manuscript. We would like to clarify that all images are entirely original and based on the discussed text. Additionally, we have included a citation in the acknowledgments to clarify the originality of the images and the software used in their creation.

A careful review of the all references is necessary. Note, for example, that the reference "Nakamura et al " (line 276) presents the wrong year and the reference on line 926 is not numbered.

Accepted and modified:

Thank you for your observation regarding the review of the references. We have identified multiple additional errors in the text and reference section, which we have addressed.

Reviewer 3 Report

Comments and Suggestions for Authors

In the current review the authors looked at the role of changes in circadian rhythm, consequent change sin cortisol production and hence adipogenesis and cancer. The authors also summarized the role of melatonin in these processes.

In general, the review is well written and comprehensive. I have a few suggestions:

1. Melatonin may be important for animals but in humans its role is limited. This limitation needs to be noted and mentioned in the review. This includes the role of melatonin as a anti-cancer molecule in humans.

2. The potential changes in IL-6, TNF. immune check point inhibitors with changes in circadian rhythm needs some discussion. Is it possible that with changes in circadian clock, the production and action of immune check point inhibitors changes and this leads to immunosuppression and so onset of cancer. The same may occur in obesity. Sine obesity is associated with immunosuppression. See: IL-17 Regulates Adipogenesis, Glucose

Eugene C. Butcher
Sofia M. Andrade, Daniel J. Cua, Fredric B. Kraemer and
Ekaterina A. Pyatnova, Andrew G. Richards, Colin Thom,
Luis A. Zúñiga, Wen-Jun Shen, Barbara Joyce-Shaikh,
http://www.jimmunol.org/content/185/11/6947
doi: 10.4049/jimmunol.1001269
October 2010;
J Immunol 20

See: n engl j med 384;12 nejm.org March 25, 2021.

See: Obesity and the tumor microenvironment Oakley C. Olson, Daniela F. Quail and Johanna A. Joyce DOI: 10.1126/science.aao5801 Science 358 (6367), 1130-1131

  3. Is there any change in the expression and function of GLP-1R in circadian alteration. 

If so, how GLP-1R agonists will be useful in such an instance as disucsse din the current review. 

Author Response

Reviewer #3

Reviewer comment

Authors response

In the current review the authors looked at the role of changes in circadian rhythm, consequent change sin cortisol production and hence adipogenesis and cancer. The authors also summarized the role of melatonin in these processes.

In general, the review is well written and comprehensive.

Thank you for your positive recognition of the comprehensive and well-structured nature of our review. Your observations have clearly helped us improve this version of the manuscript

Melatonin may be important for animals but in humans its role is limited. This limitation needs to be noted and mentioned in the review. This includes the role of melatonin as a anti-cancer molecule in humans.

Explained and modified: Thank you for your comment regarding the role of melatonin in humans compared to animal models. We would like to clarify that the importance of melatonin is unquestionable for humans. The evidence supporting its effects, particularly in modulating BMI and reducing obesity, is robust, as are its well-documented antioxidant and immunomodulatory properties. However, we agree that, compared to murine models, the clinical evidence in humans regarding its anticancer potential is more limited. To address this, we have revised two paragraphs in the perspectives section to highlight that while the emerging findings are promising, ongoing and future clinical studies will be crucial in determining the specific role or limitations of melatonin as an anticancer agent.

The potential changes in IL-6, TNF. immune check point inhibitors with changes in circadian rhythm needs some discussion. Is it possible that with changes in circadian clock, the production and action of immune check point inhibitors changes and this leads to immunosuppression and so onset of cancer. The same may occur in obesity. Sine obesity is associated with immunosuppression

Accepted and modified:

Thank you for your valuable feedback. We have expanded the discussion on the potential changes in IL-6, TNF, and immune checkpoint inhibitors in relation to circadian rhythm alterations. We explore how these changes may lead to immunosuppression and cancer onset, as well as the association between obesity and immunosuppression.

Additionally, we have included further references and enriched Figure 2 to reflect these topics more accurately.

 Is there any change in the expression and function of GLP-1R in circadian alteration. If so, how GLP-1R agonists will be useful in such an instance as disucsse din the current review. 

Accepted and modified:

Thanks for this interesting point, a paragraph discussing  this issue was added to the perspectives section.

Round 2

Reviewer 2 Report

Comments and Suggestions for Authors

I believe the manuscript is now suitable for publication since the authors have answered and/or justified the questions raised satisfactorily.

Author Response

Comment 1: I believe the manuscript is now suitable for publication since the authors have answered and/or justified the questions raised satisfactorily

Response 1: Thank you for your thorough review. We appreciate your feedback, which has strengthened our manuscript substantially. 

Reviewer 3 Report

Comments and Suggestions for Authors

The current study showing relationship between circadian rhythm on obesity and cancer is a reasonable one. This certainly needs further evaluation. In the light of this, the current review is timely and useful. But eh authors need to add a comment saying that in animals melatonin has a much significant and more direct action compared to humans. This caution is needed otherwise many scientists may think that there is indeed a direct relationship between circadian cycle and obesity and cancer.  It is also important for the authors to write a small paragraph about the relationship between circadian rhythm and response to medicines. 

Author Response

Reviewer comment

Authors response

The current study showing relationship between circadian rhythm on obesity and cancer is a reasonable one. This certainly needs further evaluation. In the light of this, the current review is timely and useful. 

We appreciate your acknowledgment of the timeliness and usefulness of this review. Based on your comment, we have added a section in the conclusions to further emphasize the need for more in-depth research in humans to clarify these associations and build upon the current findings.

The authors need to add a comment saying that in animals melatonin has a much significant and more direct action compared to humans.

We appreciate your observation. Since the first round of review, we have noted that the evidence for melatonin's action is much more abundant and significant in animals than in humans. Additionally, we discussed that in humans, circadian cycle synchronization is more influenced by social cues, whereas in animals it is predominantly governed by environmental regulation. We have revisited these ideas and added a few paragraphs to further clarify this difference.

This caution is needed otherwise many scientists may think that there is indeed a direct relationship between circadian cycle and obesity and cancer. 

Thank for your valuable recommendations. The intention of our work has not been to establish a direct relationship between circadian cycles and obesity or cancer. Rather, we have carefully outlined multiple perspectives that indirectly link circadian dysregulation with changes in cortisol or melatonin levels, leading to microenvironmental alterations that may promote the progression of various cancer types. This approach is reflected in the titles of each chapter, such as: Implications of Systemic Cortisol Deregulation in Adipogenesis, Effect of Cortisol on the Cellular Components of the TME, Melatonin Possesses Anticancer Properties, and Melatonin Can Prevent the Formation of an Adipose Microenvironment. This structure makes it clear that while the relationship is not direct, circadian dysregulation does influence conditions that can support carcinogenic processes.

Nonetheless, we have added a paragraph in the conclusion to further clarify this point.

It is also important for the authors to write a small paragraph about the relationship between circadian rhythm and response to medicines

Thank you for your valuable suggestion. We have included a comment in the perspectives section highlighting the importance of circadian health in drug responses, particularly in relation to cancer therapies. This addition emphasizes the need for further exploration of how circadian rhythms influence medication efficacy and safety.